# Room temperature electrofreezing of water yields a missing dense ice phase in the phase diagram

Weiduo Zhu[1,2,6], Yingying Huang[2,3,4,6], Chongqin Zhu[2,6], Hong-Hui Wu [2], Lu Wang[1], Jaeil Bai[2], Jinlong Yang [1], Joseph S. Francisco[2], Jijun Zhao[3], Lan-Feng Yuan[1] & Xiao Cheng Zeng [1,2,5]

Water can freeze into diverse ice polymorphs depending on the external conditions such as temperature ($T$) and pressure ($P$). Herein, molecular dynamics simulations show evidence of a high-density orthorhombic phase, termed ice χ, forming spontaneously from liquid water at room temperature under high-pressure and high external electric field. Using free-energy computations based on the Einstein molecule approach, we show that ice χ is an additional phase introduced to the state-of-the-art $T$–$P$ phase diagram. The χ phase is the most stable structure in the high-pressure/low-temperature region, located between ice II and ice VI, and next to ice V exhibiting two triple points at 6.06 kbar/131.23 K and 9.45 kbar/144.24 K, respectively. A possible explanation for the missing ice phase in the $T$–$P$ phase diagram is that ice χ is a rare polarized ferroelectric phase, whose nucleation/growth occurs only under very high electric fields.

[1] Hefei National Laboratory for Physical Sciences at Microscale, Department of Chemical Physics, University of Science and Technology of China, Hefei, Anhui 230026, China. [2] Department of Chemistry, University of Nebraska, Lincoln, NE 68588, USA. [3] Key Laboratory of Materials Modification by Laser, Ion and Electron Beams, Ministry of Education, Dalian University of Technology, Dalian 116024, China. [4] Shanghai Advanced Research Institute, Chinese Academy of Sciences, Shanghai 201210, China. [5] Department of Chemical & Biomolecular Engineering and Department of Mechanical and Materials Engineering, University of Nebraska, Lincoln, NE 68588, USA. [6] These authors contributed equally: Weiduo Zhu, Yingying Huang, Chongqin Zhu. Correspondence and requests for materials should be addressed to J.Z. (email: zhaojj@dlut.edu.cn) or to L.-F.Y. (email: yuanlf@ustc.edu.cn) or to X.C.Z. (email: xzeng1@unl.edu)

ce exhibits an exceptionally rich $T–P$ phase diagram due to the extraordinary adaptability of water's hydrogen-bonding networks to different environmental temperatures ($T$) and pressures ($P$). To date, at least 17 crystalline ice phases (ice $I_h$, $I_c$, ice II to ice XVII) have been produced in the laboratory[1–3]. A number of "computer ice" phases have also been predicted from molecular dynamics (MD) simulations and density functional theory (DFT) computations, including very-low-density porous ices (density $\rho < 0.85\,g\,cm^{-3}$) such as s-III[4], s-IV[5], ice ITT[6], and sL;[7] low-density ices ($0.85\,g\,cm^{-3} \le \rho \le 1.0\,g\,cm^{-3}$) such as silica-like ice polymorphs[8–10], ice 0[11], ice $i$, and ice $i$';[12] high-density ices ($1.0\,g\,cm^{-3} < \rho < 1.4\,g\,cm^{-3}$) such as ice B[13]; and super-high-density ices ($\rho > 2.0\,g\,cm^{-3}$), which entail partial ionization[14–20].

Among the 17 bulk ice phases observed in the laboratory, ice XI is believed to be ferroelectric[21,22] and also has been suggested to exist on Uranus and Neptune[23,24], although the recent theoretical calculations indicated that it would be antiferroelectric ice in nature[25]. The ice VIII is antiferroelectric, but it is likely to be ferroelectric in an applied electric field[26]. The ice XV, the hydrogen-ordered form of ice VI phase, is antiferroelectric ($P\bar{1}$) according to experimental observation[1], whereas it is predicted to be a ferroelectric $Cc$ hydrogen-ordered structure based on local density functional approach[27,28]. However, Del Ben et al.[29] used high-level ab initio computation and predicted that the antiferroelectric phase is indeed the ground state, suggesting that more accurate density-function approaches should be considered (see below). The ice $I_c$ is a metastable ice crystal with hydrogen disordered, but it is predicted to be ferroelectric when all hydrogens are ordered (also named as ice XIc, space group $I4_1md$) based on computer simulations[30,31]. Indeed, as the water dipole moments can add up to produce a net moment or cancel each other, either ferroelectric ice polymorphs can exist for special crystalline structures. The fabrication of bulk ferroelectric ice, however, is still a challenging task, as without the assistance of dopants as catalysts, the phase-transformation time for a single-phase ferroelectric ice is estimated to be on the order of $10^4$ years[23]. Hence, pure bulk ferroelectric ice is rare in nature.

Can a pure bulk ferroelectric ice be produced in the laboratory by other means? The answer to this question is still highly sought today. One possible way could be through the application of an ultrahigh electric field for the electrofreezing of water. Electrofreezing is known to play an important role in many natural processes, ranging from tropospheric dynamics to frost damage in cells[32–37]. In addition, several known ice structures have already been determined from computer simulations of electrofreezing under ultra-high electric fields[10,38–41]. Svishchev and Kusalik[39,41] showed from their MD simulations that a polar crystal with the structure of cubic ice $I_c$ can be achieved via electrofreezing of supercooled liquid water. In a later work, they demonstrated a MD simulation of the formation of a quartz-like ice polymorph through electrofreezing[10]. This quartz-like ice structure was originally proposed by Bernal and Fowler[42] as a type of dense ice polymorph. Stutmann[40] investigated the effects of an ultra-high electric field (tens of $V\,nm^{-1}$) on bulk water. When the electric field magnitude was $30\,V\,nm^{-1}$, the MD simulation showed that liquid water transforms into a crystal-like structure[40], which can be either defective polar cubic ice or amorphous ice. Recently, Hu et al.[38] provided simulation details on the behaviour of glassy water in external electric fields, including the formation of a body-centred-cubic (bcc) ice phase at 77 K. This bcc ice phase is polarized ferroelectric ice VIII, as determined by its lattice constant of $3.19 \pm 0.17\,\text{Å}$ and oxygen–oxygen (O–O) radial distribution function (RDF)[43,44]. However, Saitta et al.[45] showed from ab initio MD simulation that the threshold strength of electric field that makes water molecules dissociate is $3.5\,V\,nm^{-1}$. As such, the electric fields considered in aforementioned simulations were considerably higher than the threshold.

In this study, we report the formation of a previously unreported ice structure, termed ice χ, which can be observed to form spontaneously in the MD simulation of liquid water at room temperature and under an electric field below the threshold strength. The field-direction-dependent result indicates that ice χ is a rare ferroelectric phase. DFT calculations also indicate that the ice χ is dynamically stable even in zero field. Most importantly, our free-energy computation shows that ice χ is not merely a new crystalline structure but a "missing ice" phase in the contemporary $T–P$ phase diagram of ices and ice χ belongs to the family of high-density ices ($1.0\,g\,cm^{-3} < \rho < 1.4\,g\,cm^{-3}$). In the newly obtained phase diagram of ice with the TIP4P/2005 water model, ice χ is located in the high-pressure region between ice II and ice VI at low temperatures, and to the left of ice V at relatively high temperatures. The electric-field-induced crystallization of liquid water may serve as an alternative approach to attain new phase structures of water, particularly the ferroelectric ices.

## Results

**Dissociation of water under an intense electric field.** The external electric field strengths ($E$) used in our classical MD simulations (see below) are in the range of $0–3.5\,V\,nm^{-1}$. In a previous study, Saitta et al.[45] showed from ab initio MD simulation that the threshold strength of electric field that can lead to dissociation of water molecule is $3.5\,V\,nm^{-1}$. It is noteworthy that Saitta et al.[45] used the Perdew–Burke–Ernzerhof (PBE) functional and the Berry theory approach to the description of an external electric field (implemented in Quantum Expresso package). Here, instead of the PBE functional, we employed the state-of-the-art dispersion-corrected vdW-DF2 exchange-correlation functional (also called the rPW86-vdW2 functional) for the ab initio MD simulation (see Supplementary Methods for simulation details). We note that the vdW-DF2 functional has been shown to be particularly accurate for computing relative energies and transition pressures for known phases of ice (see Computational Method section). Four independent ab initio MD simulations were performed, two with the electric field of $5.0\,V\,nm^{-1}$, while another two with the electric field of $10.0\,V\,nm^{-1}$ (Supplementary Movies 1–4 and Supplementary Fig. 1). Our ab initio MD simulations show that although dissociation of water molecule can be clearly seen, within 1 ps simulation time, in bulk liquid water at 270 K and $10.0\,V\,nm^{-1}$, dissociation of water was not observed within 5 ps simulation time for the liquid water at 270 K and $5.0\,V\,nm^{-1}$; thus, such an event would be very unlikely to occur when the water was under the electric field $<3.5\,V\,nm^{-1}$.

**Spontaneous formation of ice χ under high electric field.** We performed numerous MD simulations, each with the same initial configuration of 560 TIP4P/2005 water molecules in the liquid state with the temperature controlled at 270 K (20 K above the bulk melting temperature $T_m$ of $249 \pm 2$ K[46] for the TIP4P/2005 model), while considering numerous pressures ($P$) in the range of 0.001–10 kbar (NPT ensemble). At $P = 5$ kbar and $E = 2.3\,V\,nm^{-1}$, a previously unreported ice phase emerges spontaneously, as shown in Fig. 1a, b and Supplementary Movie 5. All hydrogen atoms in solid ice χ are oriented along the $z$-axis (see Fig. 1b), i.e., the direction of the external electric field. Thus, ice χ exhibits strong polarization and can be classified as a polar ice. During the liquid-to-solid transition, the diffusion coefficient decreases sharply from $0.46 \times 10^{-5}$ to $1.38 \times 10^{-9}$ cm$^2$ s$^{-1}$. Figure 1c shows the computed O–O pair correlation function, indicating that the nearest and second nearest distances of oxygen atoms in ice χ are 0.276 and 0.334 nm, respectively. The

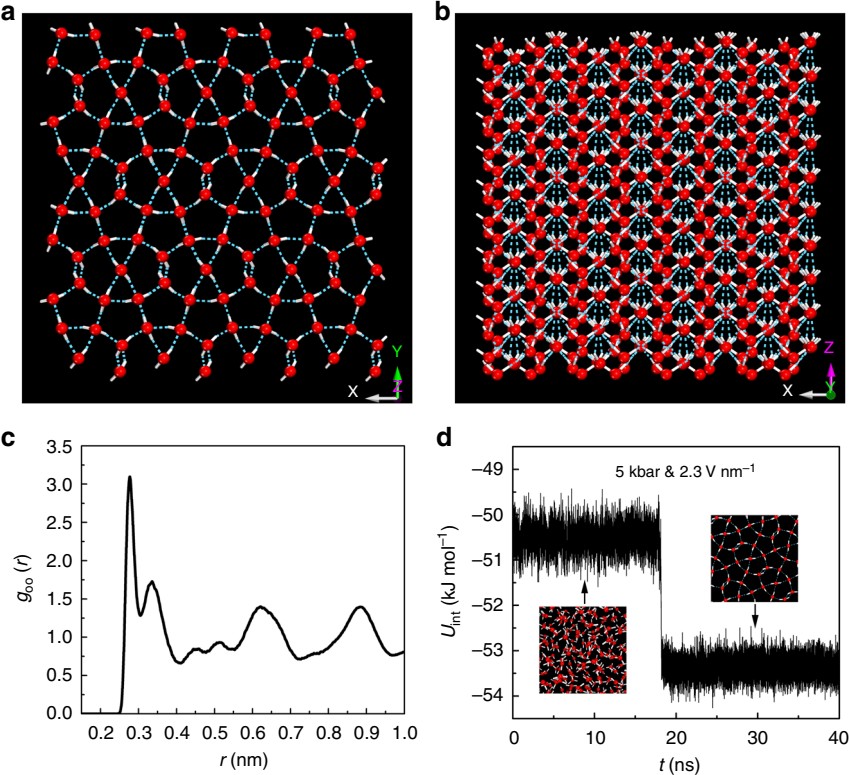

**Fig. 1** Room-temperature electrofreezing of liquid water. Snapshots of ice χ obtained at $T = 270$ K and $E = 2.3$ V nm$^{-1}$: view in the (**a**) $z$-axis and (**b**) $y$-axis direction. **c** Computed radial distribution function (RDF) of oxygen atoms for ice χ. **d** Time-dependent potential energy per water molecule for the system at $P = 5$ kbar, $T = 270$ K, and $E = 2.3$ V nm$^{-1}$. Oxygen atoms are depicted as red balls, hydrogen atoms as white sticks, and hydrogen bonds as blue dotted lines

latter distance is much shorter than that in the normal ice $I_h$ (0.495 nm)[47]. The well-separated peaks and valleys suggest that ice χ has long-range crystalline order. Figure 1d shows a notable decrease in the computed potential energy per water molecule (~ 2.9 kJ mol$^{-1}$) when the phase transformation occurs from liquid water to the crystalline ice χ, reflecting a strong first-order transition from liquid to ice χ.

As water molecules have a permanent dipole moment, in an external electric field, water molecules tend to orient along the dipole while maximizing the number of hydrogen bonds with neighbouring water molecules. Hence, in ice χ, the parallel arrangement of water dipoles is the most energetically favourable under high external electric field with the additional driving force of potential energy. The polarization energy is given by $\Delta W = E \times p$, where $E$ is the electric field and $p$ is the dipole moment per water molecule. The phase transition is favourable when the polarization energy of the water molecule is considered. Here, the threshold field strength in the electrofreezing simulation is 2.3 V nm$^{-1}$. Thus, a rough estimate of the polarization energy per water molecule is 10.2 kJ mol$^{-1}$. The total potential energy $U = U_{int} - \Delta W$ is approximately $-63.6$ kJ mol$^{-1}$, where $U_{int}$ is the interaction potential per water molecule. Compared with the initial liquid water, the potential energy (per molecule) difference is approximately $-14.0$ kJ mol$^{-1}$, in which the polarization energy accounts for ~ 73%. Therefore, the high polarization energy from the strong electric field can make the potential energy difference of the system greater than the value of $T \times \Delta S$, where $S$ is the entropy. As a result, a phase transition occurs from liquid water to ice χ. Under the electric field, the dipole orientations of water molecules tend to be along the direction of electric field, while the water molecules can

still adapt to form the hydrogen-bonding network. Their interplay could lead to a different and yet a more stable solid state.

Interestingly, our MD simulation also shows that depending on the temperature, ice χ remains stable even after switching off the external field. Under the external electric field, when the field orientation is initially along the $z$-axis and then reversed against the $z$-axis, a strong hysteresis loop is observed at pressure of 5 kbar and temperature of 200 K (see Fig. 2), confirming that ice χ is not only a polar ice but also a ferroelectric ice. In addition, the permanent electric dipole moment per water molecule is 2.211 D, close to that of a TIP4P/2005 water molecule (2.305 D)[48].

In Fig. 3a, a semi-quantitative pressure vs. electric field strength ($P$–$E$) phase diagram of the TIP4P/2005 water model at 270 K is plotted. At relatively low electric field ($E < 1.0$ V nm$^{-1}$), the water remains in the liquid state. When the electric field is high ($E > 2.5$ V nm$^{-1}$), a variety of ice polymorphs are observed, depending on the pressure of the system. The $P$–$E$ phase diagram can be divided into three regions. At $P < 3.5$ kbar, a polar ice arises, whose topological structure is the same as the previously reported (non-polar) ice B[13] (Supplementary Movie 6 and Supplementary Fig. 2). Thus, we term this ice phase polar ice B. At $3.5$ kbar $\leq P \leq 9$ kbar, liquid water transforms to solid ice χ. At $P > 9$ kbar, very-high-density amorphous (VHDA) ice is the more stable solid phase (Supplementary Fig. 3a). The intermolecular RDFs of this amorphous ice are plotted in Supplementary Fig. 3b. The first sharp peak location of amorphous ice is almost the same as that of the O–O RDF of ice χ, which denotes the nearest distance between oxygen atoms in the bulk ice. It is noteworthy that the RDF of ice χ exhibits much longer range correlation than that of the VHDA. As shown in Fig. 3b, the mass density of these ice polymorphs also increases as the pressure increases.

**Computed structural properties of ice χ based on DFT.** Next, we optimized the structure of ice χ obtained from the MD simulation using the first-principles DFT method with vdw-DF2 functional. The optimized structure is an orthorhombic crystal with space group $Fdd2$ (Supplementary Fig. 4). The lattice parameters of the unit cell are $a = 24.34$ Å, $b = 12.53$ Å, and $c = 4.32$ Å. The fractional coordinates are given in Supplementary Table 1. The top view of ice χ in the z-axis direction is displayed in Supplementary Fig. 4, where ice χ can be viewed as a column of fused oval octagons (each octagon has a centre). Based on the local surrounding environment, the 56 water molecules in the unit cell are divided into four different types ($T_1$, $T_2$, $T_3$, and $T_4$) with a population ratio of 1:2:2:2. All molecular dipoles are aligned along the z-axis, resulting in polar ice χ. As

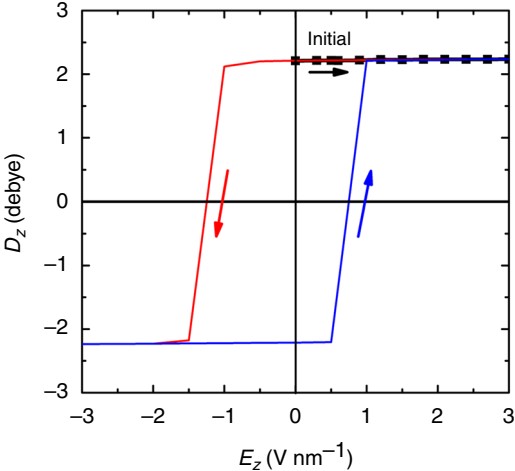

**Fig. 2** A hysteresis loop of the dipole moment per water molecule for ice χ. A hysteresis loop (blue and red lines) of the dipole moment $D_z$ per water molecule for ice χ, based on a MD simulation at $P = 5$ kbar and $T = 200$ K. The electric field $E_Z$ is applied along or against the z-axis. The black square at $E_Z = 0$ corresponds to the value of 2.211 Debye, set as the initial polarization of ice χ. The black-square line illustrates the increasing trend of electric field strength $E_Z$. The red line illustrates the decreasing trend of $E_Z$ along the z-axis until $E_Z$ reaches at zero. Thereafter, $E_Z$ increases again in the reverse direction (opposite to z-axis). The permanent electric dipole moment of a single TIP4P/2005 water molecule is 2.305 Debye

Supplementary Fig. 4d shows, the primitive cell of ice χ includes 14 water molecules and the corresponding lattice parameters are $a = 6.63$ Å, $b = 12.36$ Å, $c = 13.68$ Å, $\alpha = 28.90°$, $\beta = 64.36°$, and $\gamma = 86.74°$. The phonon dispersion and density of states are computed by using the density functional perturbation theory (DFPT) method[49], confirming dynamic stability of the ice χ (Supplementary Fig. 5).

To investigate stability competition of ice phases with different mass densities, we took the high-density ice VI[50] as a reference and considered the low-density ice XI[22], ice B[13], polar ice B, and high-density ice II[51] for the purpose of comparison. For the polar ice B obtained from our MD simulations, its fraction coordination based on DFT optimization (using the dispersion-corrected vdW-DF2 functional, also called the rPW86-vdW2 functional; see below) is given in Supplementary Table 2. The equilibrium volume of the unit cell, average nearest-neighbouring O–O distance, mass density, and lattice cohesive energy from our DFT calculation are summarized in Table 1, and were compared with available experimental data. For ice XI, the calculated mass density (0.927 g cm⁻³) is very close to the experimental value (0.93 g cm⁻³)[22] and the calculated O–O distance (2.755 Å) between neighbouring water molecules is slightly longer than the experimental value (2.735 Å)[22]. The calculated lattice cohesive energy ($E_{latt}$) of ice XI differs from the experimental value[52] by only 1.67 kJ mol⁻¹. For ice II and ice VI, their mass densities of 1.178 and 1.313 g cm⁻³, and their average O–O distances of 2.785 and 2.815 Å, obtained from our DFT calculations, are also very close to the corresponding experimental values of 1.180 and 1.310 g cm⁻³, and 2.770 and 2.800 Å, respectively[50,51]. Overall, the vdW-DF2/DFT functional reasonably describes the intermolecular hydrogen-bonding interactions of ices, as shown in our previous work[4]. For ice χ, the mass density of 1.272 g cm⁻³ is between that of ice II (1.178 g cm⁻³) and ice VI (1.313 g cm⁻³). The average O–O distance of ice χ is 2.785 Å, which is comparable to that of ice II (2.785 Å) and ice VI (2.815 Å). As shown in Table 1, for ices XI, II, χ, and VI, the lattice cohesive energy decreases with increasing mass density. However, ice B and polar ice B have lower mass densities (1.082 g cm⁻³ and 1.072 g cm⁻³) than ice II but also lower cohesive energies.

**Relative stability among various ice polymorphs at 0 K.** To compare the relative stability of ferroelectric ice χ with neighbouring ice phases in the phase diagram, we calculated their enthalpies (based on vdw-DF2 computation) under different

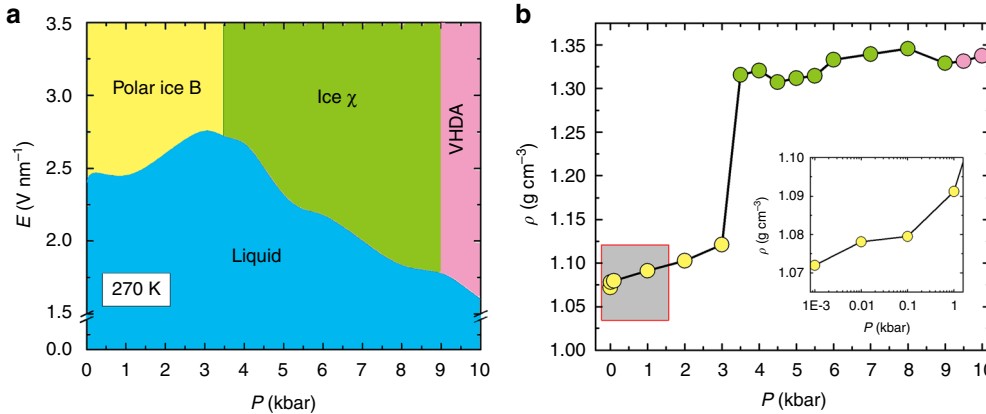

**Fig. 3** P–E phase diagram and the density of ice phases. **a** A semi-quantitative P–E phase diagram of TIP4P/2005 water for $T = 270$ K. The error bar of the electric field strength is 0.05 V nm⁻¹. Polar ice B is denoted by the yellow region, ice χ is denoted by the green region, very-high-density amorphous (VHDA) ice is denoted by the pink region, and the liquid phase is denoted by the blue region. **b** The density of the ice phase vs. P (at $T = 270$ K and $E = 3.0$ V nm⁻¹). The different colour circles correspond to different ice polymorphs, denoted by the same colour in **a**

**Table 1 Structural data on the ice polymorphs**

| Ice phase | $N_{cell}$ | $V_{cell}$ (Å³) | $d_{O-O}$ (Å) | $\rho$ (g cm⁻³) | $E_{latt}$ (kJ mol⁻¹) |
|---|---|---|---|---|---|
| Ice XI | 8 | 258.05 (257.25[a]) | 2.755 (2.735[a]) | 0.927 (0.930[a]) | 65.530 (63.86[b]) |
| Ice II | 12 | 304.65 (304.25[c]) | 2.785 (2.77[c]) | 1.178 (1.180[c]) | 65.004 (63.8[b]) |
| Ice χ | 56 | 1316.55 | 2.785 | 1.272 | 64.574 |
| Ice VI | 10 | 227.76 (227.62[d]) | 2.815 (2.81[d]) | 1.313 (1.31[d]) | 63.328 |
| Ice B | 6 | 165.91 | 2.755 | 1.082 | 64.235 |
| Polar ice B | 6 | 167.36 | 2.765 | 1.072 | 64.526 |

Number of water molecules per unit cell ($N_{cell}$), equilibrium volume of unit cell ($V_{cell}$), average distance between oxygen atoms in adjacent water molecules ($d_{O-O}$), mass density ($\rho$), and lattice cohesive energy per water molecules ($E_{latt}$). The values in parentheses are experimental data
[a]Results from neutron powder diffraction at 5 K[22]
[b]Results obtained by Whalley with zero-point energy contributions removed[52]
[c]Results from neutron diffraction at 110 K[51]
[d] Results from X-ray powder diffraction at 98 K[50]

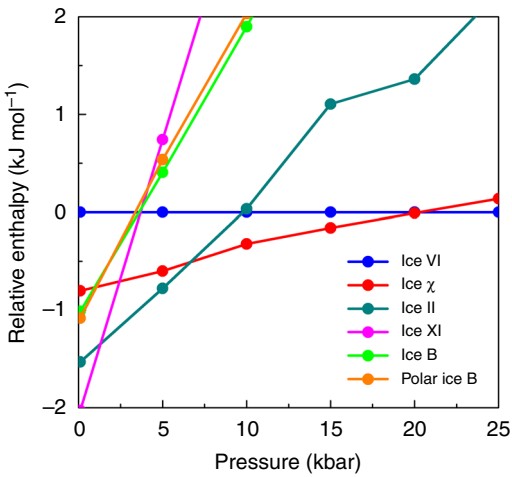

**Fig. 4** Relative enthalpy per water molecule. Relative enthalpy per water molecule (based on vdw-DF2 calculations without including ZPE correction) versus $P$ for ice χ, ice II, ice XI, ice B, and polar ice B, where ice VI is taken as the reference in the calculation

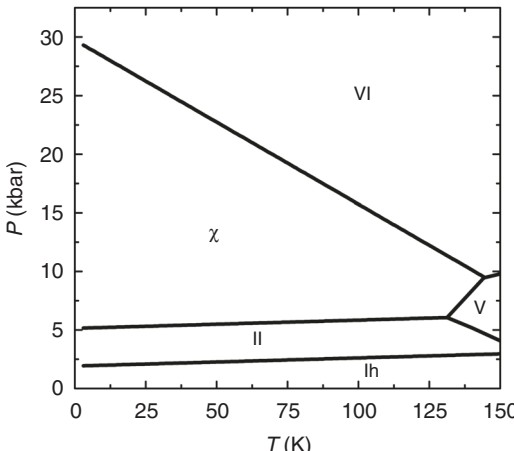

**Fig. 5** Phase diagram for TIP4P/2005 water model. The $T$–$P$ phase diagram for TIP4P/2005 water model, obtained from free-energy calculations

pressures at 0 K without including the zero-point energy correction. Figure 4 depicts the relative enthalpy (with ice VI being the reference) vs. pressure for ice XI, ice II, ice χ, ice B, and polar ice B. The point at which two curves cross marks the transition pressure between the two corresponding phases at 0 K. At 0 kbar < $P$ < 1.32 kbar, low-density ice XI is the most stable phase with the lowest enthalpy. Next, the higher-density ice II becomes more favourable at $P$ > 1.32 kbar. The transition pressure is close to the 2 kbar value that was previously obtained by Conde et al.[53] based on the TIP4P/2005 model. At $P$ > 6.66 kbar, ferroelectric ice χ replaces ice II as the most stable ice polymorph. For $P \geq 20.28$ kbar, ice VI becomes more stable than ice χ. As shown in Fig. 4, if ice χ is "missing", ice II would transform directly into ice VI at $P = 9.79$ kbar, consistent with the transition pressure of 10 kbar obtained by Conde et al.[53] Although neither is the most stable phase, as indicated by the enthalpy curves, ice B and polar ice B have nearly the same stability. Again, our DFT calculations demonstrate that ice χ is one of the most stable high-density ices in the high-pressure region at zero temperature, along with the known phases of ice II and ice VI. In addition, we calculated enthalpies of these ice structures at different pressures and 0 K, using the strongly constrained and appropriately normed[54] functional (Supplementary Fig. 6). The results are consistent with vdW-DF2/DFT computation, demonstrating that ice χ is a highly stable high-density ice in the high-pressure region and at zero temperature (more details see Supplementary Information).

**Free-energy computation of $T$–$P$ phase diagram.** Lastly, to examine the stability of ice χ at temperatures much higher than 0 K, we performed free-energy calculations using the Einstein molecule approach method. In our previous study, we confirmed that the TIP4P/2005 water model can reasonably simulate the realistic $T$–$P$ phase diagram of water/ice[4]. Aragones et al.[55,56] also showed that the TIP4P/2005 water model can describe the relative energy, critical temperature, and surface tension of liquid water and ice phases well. The $T$–$P$ phase diagram of water/ice is plotted in Fig. 5. Four ice polymorphs, namely, ice $I_h$ (or hydrogen-disordered ice XI), ice II, ferroelectric ice χ, and ice VI, arise in sequence with increasing pressure at low temperature. Extrapolation of the phase boundaries to 0 K gives the corresponding transition pressures of 1.92, 5.14, and 29.73 kbar, respectively, compared with 1.32, 6.66, and 20.28 kbar predicted from the above DFT computations. The predicted transition pressure of 1.92 kbar at 0 K for ice $I_h$ to ice II is in excellent agreement with that of the 2 kbar value previously obtained by Conde et al.[53] Different from the previous $T$–$P$ phase diagram, the coexistent line between ice II and ice VI disappears, whereas ferroelectric ice χ occupies a region between ice II and ice VI at low temperature and part of the region of ice V at relatively high temperature. As a result, two new triple points emerge: one for ices II, V, and χ at 6.06 kbar and 131.23 K, and the other for ices χ, V, and VI at 9.45 kbar and 144.24 K. Ice B and polar ice B do not appear in the $T$–$P$ diagram, as they have higher Gibbs free energies. It is noteworthy that the free-energy calculations show

that ice χ has the lowest free energy among ice χ, ice XIII, and ice XV in the low-temperature and high-pressure region, indicating that ice χ is more stable than ice XIII and ice XV. Overall, both MD simulations at finite temperature and DFT calculations at 0 K strongly support the existence of ferroelectric ice χ in the $T–P$ phase diagram at high pressures.

## Discussion

We predict a new ferroelectric ice χ in the phase diagram of water. Ferroelectric ice χ has a high mass density of $1.27\,\mathrm{g\,cm^{-3}}$. The ferroelectric ice χ is also proven to be dynamically stable on the basis of phonon-spectrum DFT computation. In the $T–P$ phase diagram of water/ice, ferroelectric ice χ emerges in the high-pressure region, located between ice II and ice VI at low temperatures and occupying some domains of ice V at relatively high temperatures, leading to two triple points at $P = 6.06$ kbar and $T = 131.23$ K, and at $P = 9.45$ kbar and $T = 144.24$ K, respectively. The appearance of ice χ in the $T–P$ phase diagram of water/ice suggests that the ferroelectric ice χ entails high thermodynamic stability. Identification of this ferroelectric ice phase not only reveals a "missing" ice polymorph in the high-pressure region of the state-of-the-art $T–P$ phase diagram of water but also provides more precise temperature/pressure conditions for seeking the elusive ferroelectric ice. In light of the requirement of ultrahigh electric field, whether this predicted ice χ can be produced in the laboratory via electrofreezing of liquid water remains to be an open question.

## Methods

**MD simulations**. All MD simulations are performed in the isothermal-isobaric ($NPT$) ensemble, with an external electric field applied along the $z$-axis. The MD simulations are undertaken with the GROMACS (GROningen MAchine for Chemical Simulations) package[57]. For all MD simulations, the initial supercell is a cubic box containing 560 TIP4P/2005 water molecules[48] and the temperature is controlled at 270 K (corresponding to room temperature for the TIP4P/2005 model, as the melting point of the bulk ice $I_h$ based on the TIP4P/2005 model is approximately 250 K)[46]. To map out a semi-quantitative $P–E$ phase diagram, we set a series of pressure and electric field values, including $P = 0.001, 0.01, 0.05, 0.1, 1.0, 2.0, 3.0, 3.5, 4.0, 4.5, 5.0, 6.0, 7.0, 8.0, 9.0, 9.5,$ and $10.0$ kbar, whereas the electric field strength is varied from 0 to $3.5\,\mathrm{V\,nm^{-1}}$ by an increment of $0.1\,\mathrm{V\,nm^{-1}}$. The external electric field induced an additional force, $F_i = q_i E$, where $F_i$ is the force induced by the electric field, $q_i$ is the charge of each atom, and $E$ is the applied electric field. For all simulations, the time step is set to 2 fs. The equilibration MD run lasted for at least 10 ns and, in some cases, lasted for 200 ns. $T$ and $P$ are controlled by a Nosé–Hoover thermostat[58] and a Parrinello–Rahman barostat[59], respectively. A cutoff of 1.0 nm is adopted for the $L–J$ interactions and the long-range electrostatic interactions are treated by the slab-adapted Ewald sum method[60].

**DFT calculations**. The relative energies of ice χ and selected ice phases are computed using the DFT methods implemented in the VASP 5.3.5 software package[61]. The electron-ion interactions are described by the projector augmented wave potential[62]. To account for the intermolecular dispersion interactions, the exchange-correlation interaction is described by the dispersion corrected vdW-DF2 exchange-correlation functional[63] (also called the rPW86-vdW2 functional). We note that the vdw-DF2 functional has been previously shown by Santra et al.[61] to be particularly accurate for computing relative energies and transition pressures for known phases of ices. The electron wavefunction is expanded by a plane-wave basis up to 700 eV. The Brillouin zones are sampled by $k$-point grids with a uniform spacing of $2\pi \times 0.04\,\text{Å}^{-1}$. To confirm dynamic stability of ice χ, the phonon dispersion is computed by using the DFPT method[49] as implemented in the VASP 5.4.

**Monte Carlo/MD simulations of the $T–P$ phase diagram**. The $T–P$ phase diagram of water and ice polymorphs (including ice $I_h$, ice II, ice V, ice VI, hydrogen-ordered ice XIII and ice XV, previously predicted polar ice B, and predicted ice χ from this study) was derived based on the Einstein model for crystals and the TIP4P/2005 water potential. First, to obtain reliable configurations of the ice polymorphs, isothermal-isobaric Monte Carlo simulations at temperatures from 1 to 200 K (with 10 K increments) and pressures from 1 to 24 kbar (with 1 kbar increments) are performed using a homemade code. For each candidate phase, the configurations from the Monte Carlo simulations are used to calculate the free energy on the basis of the Einstein molecule approach with the GROMACS programme[57]. At each $T–P$ condition, the Ewald sum method with a real-space cutoff of 8.5 Å is adopted to treat the electrostatic interactions and the pair potential is truncated at 8.5 Å. For ice $I_h$, ice V, and ice VI, the effect of hydrogen disorder is considered in the free-energy

computation and their Pauling entropies $S/Nk_B$ are taken as $\ln(3/2)$, 0.3817, and $\ln(3/2)$, respectively[64]. Once the free energy at a reference point is determined, the thermodynamic integration method can be used to evaluate the free energy under other thermodynamic conditions. Specifically, an initial coexistent point is located by equating the chemical potentials of two phases at a given temperature and pressure[65]. Next, the Gibbs–Duhem integration based on the trapezoid predictor-corrector formulas is performed to compute the phase boundaries[66].

## Data availability
The authors declare that the data and code related to this study are available upon reasonable request.

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

## Acknowledgements

We gratefully acknowledge the financial support from the US National Science Foundation (CHE-1665325), UNL Nebraska Center for Energy Sciences Research, CNPC and CAS (2015A-4812), the National Natural Science Foundation of China (11574282), CAS Strategic Priority Research Program (XDB10030402), and the National Key Research & Development Program of China (Grant No. 2016YFA0200604). The computational work was performed at the University of Nebraska Holland Computing Center.

## Author contributions

X.C.Z. initiated the idea. X.C.Z., J.Z. and L.-F.Y. supervised the research. W.Z., Y.H. and C.Z. carried out the calculations and analysis with the help from H.W. and L.W. C.Z. provided the initial code of the free-energy calculations. W.Z., Y.H. and C.Z. contributed equally. J.B., J.Y. and J.S.F. provided tools of analysis and valuable discussions. W.Z. and Y.H. prepared the initial draft of the manuscript. All authors contributed to the discussions and revisions of the manuscript.

## Additional information

**Competing interests:** The authors declare no competing interests.

**Journal Peer Review Information**: *Nature Communications* thanks Edgar Engel and other anonymous reviewers for their contribution to the peer review of this work. Peer reviewer reports are available.

