## [Peer Review File · Nature Communications]

Reviewers' comments:

Reviewer #1 (Remarks to the Author):

The authors report the theoretical discovery of a new high-pressure phase of water ice, which they predict to be stable at ambient temperatures and experimentally-realizable pressures and whose synthesis is facilitated substantially by application of an external electric field.

In some ways their work follows a series of publications proposing (individual) new theoretical ice phases, such as (JCP 147, 091101 (2017)) and (PCCP 20, 8333 (2018)).

Their work somewhat distinguishes itself from the typical theoretical prediction and ice phases by (1) considering a stabilizing mechanism other than pressure (external electric fields) and (2) identifying a new phase from explicit simulation of its nucleation using MD methods.

While neither stabilization of ice phases by electric fields (as acknowledged by the authors in their introduction) nor plain MD simulations of nucleation processes are revolutionary, the result is an ice phase, which – to my knowledge – has not been discussed in literature, and which represents a substantial challenge to first-principles structure searching approaches due to its large unit cell size.

Proposals of individual novel theoretical ice structures are not particularly rare, but have recently largely focussed on low-pressure clathrate ice structures which are computationally stable under severe negative pressures and without clear prospects for experimental synthesis (protocols). The fact that the authors provide explicit estimates of the regime of stability in terms of pressure (and temperature) and suggest electric field required to facilitate synthesis provides a seemingly viable and realizable starting point for experiments and probably renders the work of higher interest to the ice community than the typical proposal of a novel ice structure.

However, this brings me to some important weaknesses of this work.

Firstly, not just the stability regime but also the more fundamental statement regarding the zero- or ambient-temperature and zero electric field stability of the proposed phase require much more support. Both the TIP4P/2005 force field used to simulate the nucleation of the proposed structure subject to external electric fields and to construct the P-T diagram, and the vdW-DF2 functional employed to in the first-principles calculation of the pressure dependence of the relative stabilities of the proposed ice phase and different known phases are well-established. TIP4P/2005 has been shown to reproduce the phase diagram of ice as we know it surprisingly well, however the vdW-DF2 functional (although by no means a bad choice) is not the canonical functional to choose for ice. The rPW86-vdW2 and (more recently) SCAN functional are probably to two best choices (JCP 139, 154702 (2013), JCP 144, 130901 (2016), and Nature Chemistry 8, 831 (2016)) and should be considered here considering that differences in stabilities of less than 0.5 kJ/mol are under discussion here.

Secondly, I suspect that the very limited characterization of the phase (which would allow to identify synthesis experimentally) will be a thorn in the eyes of the experimental community. IR/Rahman spectra should be possible to calculate at little additional effort and come essentially for free when performing vibrational analyses (which I will get to in a moment). Proton momentum distributions would probably much appreciated as well but require substantially more work, since quantum treatment of nuclear motion is required.

Thirdly, effects of quantum nuclear motion at least at the harmonic level, but better at the quasi-harmonic or even anharmonic level should be considered in view of the subtle relative stability of the proposed field at zero electric field of less than 0.5 kJ/mol, since (harmonic and anharmonic) vibrational contributions to the free energy can easily differ by amounts comparable to that number and be crucial in determining phase stability (e.g. JCP 139, 084503 (2013) and PRX 5,

021033 (2015)).

In conclusion, the work is unlikely to result in a shift of paradigm in either computational structure searching and materials design or our understanding of nature of water and ice under high pressures, but should garner the interest of both the experimental and theoretical/computational ice communities.

In my view the manuscript is not yet acceptable for publication in its present form, but would be substantially strengthened by providing more support that the results are method-insensitive rather than artefacts of the particular levels of theory employed in the study. I suspect that an experimental reviewer would request calculation of spectroscopic properties suitable for experimental identification of the structure.

More technical comments are the following:

1. Since the abstract provides specific predictions for triple points it should mention the level of theory at which these were calculated.
2. The list of low-density porous ices and silica-like ice polymorphs provided in the introduction should probably include those proposed in PCCP 20, 8333 (2018) and JCP 147, 091101 (2017).
3. In the introduction the authors claim that only ice XI is believed to be ferroelectric, which is a strong claim considering evidence for ferroelectric ice Xic (e.g. PCCP 13, 19788 (2011) and PRX 5, 021033 (2015) and references therein) and VIII (F. Okada and K. Naya, Electrolysis for ozone water production, In Electrolysis, Ed. V. Linkov (2012) pp 243-272).
4. In multiple instances the authors use oxygen-oxygen distances as a measure of density, which they are not.
5. Since the authors first discuss results obtained the TIP4P/2005 force field, then vdW-DF2 density-functional-theory, and then once again the TIP4P/2005 force field, they should make more clear which level of theory different numbers (such as the potential-energies on p. 5) were calculated with.
6. If possible it should be substantiated that electric fields of around 2.5 V/nm as proposed to facilitate synthesis of the proposed phase are experimentally-realizable.
7. It would be useful to be able to compare the RDFs of the proposed phase and e.g. VHDA. The authors should consider incorporating the RDF in Fig. S2 (b) in Fig. 1 (c).
8. It should be made quite clear that the "relative stability from DFT calculations at 0 K" is really a static-lattice stability from DFT calculations, since (if I am not mistaken) zero-point nuclear motion was not considered (and is not unlikely to play an important role).
9. Table 1 and Figs. 4 and 5 should include ices V, VII and VIII, ice V, and ices VII and VIII, respectively. For example in Fig. 5 ices VII and VIII are expected to appear on the shown TIP4P/2005 P-T phase diagram according to JCP 130, 244504 (2009).
10. The conclusions unnecessarily reiterate elements of the results section.

Reviewer #2 (Remarks to the Author):

The manuscript by Zhu and coworkers contains an interesting computational study that suggests that a previously unknown phase of ice forms in the presence of a very strong electric field and high pressure. These are very interesting findings that are worth publishing in Nat. Comm. However, it should also be acknowledged that water / ice will turn into oxygen and hydrogen gas under such extreme voltages and realistic electrode separations in any experiment. This is obviously a chemical "degree of freedom" not considered in these calculations. Also, the authors need to address the following points before publication:

* The new phase must not be called "ice XX". Roman numerals are reserved for new experimentally-found phases. According to Petrenko's Physics and Chemistry of Ice book, experimentalists must provide crystallographic or at least spectroscopic evidence in order to use a new Roman numeral. Perhaps "polar ice C" would be an adequate name for the new ice?

* page 3, line 36: Seventeen phases have been found so far. The members of the ice I family are polytypes and thus only count as one phase.

* page 3, line 47: The authors give the definition of a ferroelectric phase. Strictly speaking, we only know that ice XI is pyroelectric. The reversal of polarity with an external electric field has not been conclusively demonstrated.

* page 3, line 49: The term "proton-ordered" is chemically incorrect. The water molecule consists of hydrogen and oxygen atoms. The hydrogen atoms are ordered - hence "hydrogen-ordered" is the correct term. There is confusion about this in the literature, but it is important to get this right since excess protons (H⁺) can induce hydrogen ordering. In this context, it should be mentioned that the ferroelectricity of ice XI is controversial. Please see: J. Phys. Chem. C 2014, 118, 26264–26275

* page 3, line 62: It needs to be said that this work is computational.

* page 4, line 77: Reading this paragraph, it is not clear to me if the authors suggest that the new phase is also stable in the absence of an electric field or not? This needs to be clarified. In general, this paragraph should not be a summary of the work but an outlook to what the work tries to achieve building on the previous work described in the introduction.

* page 4, line 84: The definition of "very-high-density" seems somewhat random to me. I would simply refer to these ices as high-density ices. There are a range of phases that occur around the indicated density value, so some would rather randomly fall into the category and others not. If anything, I would use the interpenetration of hydrogen bonds as a definition for "very-high-density". I don't think this is observed for the new ice?

* page 5, line 97: It needs to be mentioned that 2.3 V/nm is a very strong electric field. Making the ice/water layer thicker than a few nanometers would result immediately in the chemical splitting of water into hydrogen and oxygen gas while going smaller means that the phase behaviour is dictated by the nanodimensions. Experimentally, in my judgement, it would be impossible (by several orders of magnitude) to realise the conditions where the new phase forms. In Figure 1(a,b) the unit cell should be indicated.

* page 5, line 100: What is the spacegroup of the new ice? This will reflect its polar nature. So it is relevant at this stage of the manuscript.

* page 6, line 127: The rotational degrees of freedom are restricted in all phases of ice due to the hydrogen bonding. The authors need to make their point clearer here.

* page 6, line 136: It needs to be stated if the new ice "melts" during the hysteresis test or not. If it melts, then it would not qualify as a ferroelectric phase.

* Figure 3(a): How was the phase diagram constructed? Are the "wiggly" phase boundaries real?

* Figure 4: Does it not make sense to only use hydrogen ordered phases for comparison here? Are the authors sure they used ice VI and not ice XV? If they used ice VI, how was the hydrogen disorder implemented and represented?

* page 10, line 241: Only TIP4P was tested for the DFT calculations. Other functionals should be tested as well. It should also be acknowledged that DFT generally favours ferroelectric structure due to the tin-foil approximation. Please see: J. Phys. Chem. Lett. 2014, 5, 4122–4128

* page 10, line 254: Again, it needs to be acknowledged that the experimental preparation of the new ice would not be possible since the water would chemically decompose into oxygen and hydrogen gas. Also, I am still a bit confused if the authors suggest that the new ice would be stable in absence of an electric field once formed? The phase diagram in Figure 5 seems to suggest that it is thermodynamically stable in the absence of an electric field. However, in the last paragraph it is also said that the new ice is only "dynamically stable". These points are obviously key and need to be spelled out very clearly.

* page 16, Table 1: In-situ values for the densities of ices II and VI are available in the literature and should be used here. D. Fortes has published in-situ data for both phases.

Reviewer #3 (Remarks to the Author):

Dear Editor

I have read the manuscript titled "Room-temperature electrofreezing of liquid water: a "missing" dense ice phase in the phase diagram" by Zhu and coworkers.

In their work, the authors perform a numerical study of water under an applied high electric field and high pressure and propose a new ice form, ice XX, presented as the most stable structure in the high pressure/low temperature region. According to the authors, ice XX is a polarised ferroelectric phase, whose nucleation occurs only under these extreme conditions.

I regret to state that the manuscript is not well written, missing not only the clearness of the language but also too much information necessary to be able to reproduce the presented results, thus not deserving publication in Nature Communication.

In what follows, I will give few reasons to support my choice.

1- The authors seem to suggest they have discovered a new ice phase, that should appear in the experimental phase diagram under an applied high electric field of few V/nm. However, they never comment on the fact that this value of the electric field is far beyond the dielectric strength of real water. Therefore, one could argue that this phase would never appear in nature.

2- In their study, the authors use TIP4P/2005 water but never discuss the fact that for this water model, the dielectric constants of liquid water and of ice have the opposite behaviour with respect to their real counterparts.

3- The authors vary parameters such as pressure and electric field, not knowing the underlying phase diagram of the water model, and not even properly computing it.

4- The authors study the stability of the novel ice at 0K by means of enthalpy. Next, they establish

that this ice should be located between ice II and VI between 0K and 120K. However, when presenting the free-energy calculations, not only the numerical procedure is not well explained and numerical tables are missing, but also the Pauling entropy is not even mentioned, even though its contribution on the stability of the proposed ice might play an important role and should have been discussed by the authors.

Point-by-Point Responses to the Comments

Reviewer #1 (Remarks to the Author):

The authors report the theoretical discovery of a new high-pressure phase of water ice, which they predict to be stable at ambient temperatures and experimentally-realizable pressures and whose synthesis is facilitated substantially by application of an external electric field.

In some ways their work follows a series of publications proposing (individual) new theoretical ice phases, such as (JCP 147, 091101 (2017)) and (PCCP 20, 8333 (2018)). Their work somewhat distinguishes itself from the typical theoretical prediction and ice phases by (1) considering a stabilizing mechanism other than pressure (external electric fields) and (2) identifying a new phase from explicit simulation of its nucleation using MD methods.

While neither stabilization of ice phases by electric fields (as acknowledged by the authors in their introduction) nor plain MD simulations of nucleation processes are revolutionary, **the result is an ice phase, which – to my knowledge – has not been discussed in literature,** and which represents a substantial challenge to first-principles structure searching approaches due to its large unit cell size.

Proposals of individual novel theoretical ice structures are not particularly rare, but have recently largely focussed on low-pressure clathrate ice structures which are computationally stable under severe negative pressures and without clear prospects for experimental synthesis (protocols).

The fact that the authors provide explicit estimates of the regime of stability in terms of pressure (and temperature) and suggest **electric field required to facilitate synthesis provides a seemingly viable and realizable starting point for experiments and probably renders the work of higher interest to the ice community than the typical proposal of a novel ice structure.**

Response: We thank the Review #1 for the positive comments, especially for pointing out “renders the work of higher interest to the ice community than the typical proposal of a novel ice structure.”

However, this brings me to some important weaknesses of this work.

Firstly, not just the stability regime but also the more fundamental statement regarding the zero- or ambient-temperature and zero electric field stability of the proposed phase require much more support. Both the TIP4P/2005 force field used to simulate the nucleation of the proposed structure subject to external electric fields and to construct the P-T diagram, and the vdW-DF2 functional employed to in the first-principles calculation of the pressure dependence of the relative stabilities of the proposed ice

phase and different known phases are well-established. TIP4P/2005 has been shown to reproduce the phase diagram of ice as we know it surprisingly well, however the vdW-DF2 functional (although by no means a bad choice) is not the canonical functional to choose for ice. The rPW86-vdW2 and (more recently) SCAN functional are probably two of the best choices (JCP 139, 154702 (2013), JCP 144, 130901 (2016), and Nature Chemistry 8, 831 (2016)) and should be considered here considering that differences in stabilities of less than 0.5 kJ/mol are under discussion here.

Response: Thank for the valuable suggestion. After literature search, we find that the vdw-DF2 functional is the same as rPW86-vdW2 functional. It is interesting the same functional has two names. Indeed, the rPW86-vdW2 functional is one of the two best choices, as our calculation results are very close to the experimental ones, e.g., the density and oxygen-oxygen distance of ices (see Table 1).

We also add new results with using the second best SCAN functional (see Supplementary results). The results are consistent with vdW-DF2 computations, demonstrating that ice χ is one of the most stable very-high-density ices in the high-pressure region at zero temperature.

Secondly, I suspect that the very limited characterization of the phase (which would allow to identify synthesis experimentally) will be a thorn in the eyes of the experimental community. IR/Rahman spectra should be possible to calculate at little additional effort and come essentially for free when performing vibrational analyses (which I will get to in a moment). Proton momentum distributions would probably be much appreciated as well but require substantially more work, since quantum treatment of nuclear motion is required.

Response: These are very good suggestions. The infrared spectrum is now added in the Supplementary Information (see Figure S4b). Due to the expensive proton momentum distribution calculations, we will need to invest more resources on this computation in future.

Thirdly, effects of quantum nuclear motion at least at the harmonic level, but better at the quasi-harmonic or even anharmonic level should be considered in view of the subtle relative stability of the proposed field at zero electric field of less than 0.5 kJ/mol, since (harmonic and anharmonic) vibrational contributions to the free energy can easily differ by amounts comparable to that number and be crucial in determining phase stability (e.g. JCP 139, 084503 (2013) and PRX 5, 021033 (2015)).

Response: Thanks for the suggestion. We have calculated the zero-point energy (ZPE) correction at the harmonic level to modify the free energy. The results are shown in Figure R1 below. Our results indicate that the effects of quantum nuclear motion do not affect the relative stability of those ice phases, but just the stability region between them. As shown in Figure R1(a) and (b). The ice χ still appears in the high pressure

region at the most stable phase between ice II and ice VI.

Figure R1. Relative enthalpy per water molecule (based on DFT calculations) versus P for ice χ , ice II, ice XI, ice B, and polar ice B, where ice VI is taken as the reference in the calculation. (a) Without ZPE and (b) with ZPE correction.

In conclusion, the work is unlikely to result in a shift of paradigm in either computational structure searching and materials design or our understanding of nature of water and ice under high pressures, **but should garner the interest of both the experimental and theoretical/computational ice communities.**

In my view the manuscript is not yet acceptable for publication in its present form, but would be substantially strengthened by providing more support that the results are method-insensitive rather than artefacts of the particular levels of theory employed in the study. I suspect that an experimental reviewer would request calculation of spectroscopic properties suitable for experimental identification of the structure.

More technical comments are the following:

1. Since the abstract provides specific predictions for triple points it should mention the level of theory at which these were calculated.

Response: vdw-DF2 or rPW86-vdW2_functional was used for optimization. Classical MD simulations and free-energy calculations were performed with TIP4P/2005 water model. We made these information clearer in the revised manuscript.

2. The list of low-density porous ices and silica-like ice polymorphs provided in the introduction should probably include those proposed in PCCP 20, 8333 (2018) and JCP 147, 091101 (2017).

Response: We have added both references in the revised manuscript.

3. In the introduction the authors claim that only ice XI is believed to be ferroelectric,

which is a strong claim considering evidence for ferroelectric ice Xic (e.g. PCCP 13, 19788 (2011) and PRX 5, 021033 (2015) and references therein) and VIII (F. Okada and K. Naya, Electrolysis for ozone water production, In Electrolysis, Ed. V. Linkov (2012) pp 243-272).

Response: Thanks for the good point and references. We have revised the statement. The ice Xic is only predicted to be ferroelectric by theoretical calculations, but has not been confirmed by experiment. And the ice VIII found in experiment is believed to be antiferroelectric³⁻⁵. It is possible that a ferroelectric variant of this structure may exist in an applied electric field, but the antiferroelectric structure is more stable than that ferroelectric structure. In phase diagram of water, only ice XI is believed to be ferroelectric and confirmed by neutron diffraction (See refs. 14, 15 below).

4. In multiple instances the authors use oxygen-oxygen distances as a measure of density, which they are not.

Response: Thanks for pointing out this issue. We have corrected the statements in the revised manuscript.

5. Since the authors first discuss results obtained the TIP4P/2005 force field, then vdW-DF2 density-functional-theory, and then once again the TIP4P/2005 force field, they should make more clear which level of theory different numbers (such as the potential-energies on p. 5) were calculated with.

Response: Thanks for the suggestion. We have clarified the level of theory used, in the revised manuscript.

6. If possible it should be substantiated that electric fields of around 2.5 V/nm as proposed to facilitate synthesis of the proposed phase are experimentally-realizable.

Response: Thanks for the comment. We found a report that electric field as high as 40 V/nm, which is much stronger than that required to facilitate synthesis of the proposed ice, can be experimentally realizable⁶.

7. It would be useful to be able to compare the RDFs of the proposed phase and e.g. VHDA. The authors should consider incorporating the RDF in Fig. S2 (b) in Fig. 1 (c).

Response: Thanks for the suggestion. Because the discussion part of Fig. 1 focuses on the new ice structure, we still like to keep RDF of VHDA in Fig. S2(b), following the order that we introduce different phases.

8. It should be made quite clear that the “relative stability from DFT calculations at 0 K” is really a static-lattice stability from DFT calculations, since (if I am not mistaken) zero-point nuclear motion was not considered (and is not unlikely to play an important role).

Response: We have added the contribution of zero point vibrational energy, and recalculated the lattice cohesive energy of ice XI, ice II, ice XX and ice VI. The values of lattice cohesive energy are 49.900 kJ/mol, 49.503 kJ/mol, 48.843 kJ/mol

and 48.237 kJ/mol, respectively. The relative difference does not change qualitatively.

9. Table 1 and Figs. 4 and 5 should include ices V, VII and VIII, ice V, and ices VII and VIII, respectively. For example in Fig. 5 ices VII and VIII are expected to appear on the shown TIP4P/2005 P-T phase diagram according to JCP 130, 244504 (2009).

Response: Thanks for the comment. We have calculated the free energy of ice V, ice VII and ice VIII with TIP4P/2005 water model, obtained similar results as the JCP paper cited by the Reviewer. Here, we mainly focus on the proposed new ice phase as the most stable structure in the high-pressure/low-temperature region. For clarity, we have not plotted ice VII and VIII in the phase diagram. We add the citation of (JCP 130, 244504 (2009), see ref. 55) in the revised manuscript.

10. The conclusions unnecessarily reiterate elements of the results section.

Response: We have made changes in the revised manuscript.

Reviewer #2 (Remarks to the Author):

The manuscript by Zhu and coworkers contains an interesting computational study that suggests that a previously unknown phase of ice forms in the presence of a very strong electric field and high pressure. **These are very interesting findings that are worth publishing in Nat. Comm.** However, it should also be acknowledged that water / ice will turn into oxygen and hydrogen gas under such extreme voltages and realistic electrode separations in any experiment. This is obviously a chemical "degree of freedom" not considered in these calculations. Also, the authors need to address the following points before publication:

Response: Thank the Referee for the positive comments, especially for pointing out "worth publishing in Nat. Comm."

* The new phase must not be called "ice XX". Roman numerals are reserved for new experimentally-found phases. According to Petrenko's Physics and Chemistry of Ice book, experimentalists must provide crystallographic or at least spectroscopic evidence in order to use a new Roman numeral. Perhaps "polar ice C" would be an adequate name for the new ice?

Response: This is a valid point. We re-name this new ice phase with a Greek letter χ .

* page 3, line 36: Seventeen phases have been found so far. The members of the ice I family are polytypes and thus only count as one phase.

Response: The point is well taken. We changed 18 to 17.

* page 3, line 47: The authors give the definition of a ferroelectric phase. Strictly

speaking, we only know that ice XI is pyroelectric. The reversal of polarity with an external electric field has not been conclusively demonstrated.

Response: Thanks for the comment. Fukazawa et al. did neutron diffraction experiment^{14, 15} and suggested that the ice XI is ferroelectric. If reviewer #2 still thinks that ice XI is pyroelectric due to the lack of reversal of polarity, we can change the term back.

* page 3, line 49: The term "proton-ordered" is chemically incorrect. The water molecule consists of hydrogen and oxygen atoms. The hydrogen atoms are ordered - hence "hydrogen-ordered" is the correct term. There is confusion about this in the literature, but it is important to get this right since excess protons (H⁺) can induce hydrogen ordering. In this context, it should be mentioned that the ferroelectricity of ice XI is controversial. Please see: J. Phys. Chem. C 2014, 118, 26264–26275

Response: We noticed numerous groups have used the terms proton-order and proton-disorder to describe the states of hydrogen atoms in ice crystals, especially those ice phases determined by experiment¹⁶⁻²⁰. Again, if reviewer #2 felt that these usages are confusing, we could change the term to hydrogen-ordered. We are aware of the theoretical paper pointed out by the reviewer.

* page 3, line 62: It needs to be said that this work is computational.

Response: We have made the change in the revised manuscript.

* page 4, line 77: Reading this paragraph, it is not clear to me if the authors suggest that the new phase is also stable in the absence of an electric field or not? This needs to be clarified. In general, this paragraph should not be a summary of the work but an outlook to what the work tries to achieve building on the previous work described in the introduction.

Response: Please note that once formed, the ice χ is stable even the field is turned off at low temperature.

* page 4, line 84: The definition of "very-high-density" seems somewhat random to me. I would simply refer to these ices as high-density ices. There are a range of phases that occur around the indicated density value, so some would rather randomly fall into the category and others not. If anything, I would use the interpenetration of hydrogen bonds as a definition for "very-high-density". I don't think this is observed for the new ice?

Response: Thanks for the comment. We were trying to use the density term to be consistent with previous studies^{21, 22}. Previously, the amorphous ices are classified into three types based on their density: low density amorphous (LDA) ice ($\rho = 0.925 \text{ g/cm}^3$), high density amorphous ice (HDA) ($\rho = 1.17 \text{ g/cm}^3$), and very high density amorphous (VHDA) ice ($\rho = 1.25 \text{ g/cm}^3$). Here, since the density of new ice χ is almost the same as that of VHDA, we call it VHDI to be consistent with the widely used term in the literature.

* page 5, line 97: It needs to be mentioned that 2.3 V/nm is a very strong electric field. Making the ice/water layer thicker than a few nanometers would result immediately in the chemical splitting of water into hydrogen and oxygen gas while going smaller means that the phase behaviour is dictated by the nanodimensions. Experimentally, in my judgement, it would be impossible (by several orders of magnitude) to realise the conditions where the new phase forms. In Figure 1(a,b) the unit cell should be indicated.

Response:

Thanks for the comment. We note that in the previous work by Saitta *et al.*, they indicated that electric fields beyond a threshold of about 3.5 V/nm can dissociate water molecules¹¹. So in our work, we adopt an electric field < 3.5 V/nm.

Note also that we use a relatively high electric field in MD simulation to speed up the ice formation due to the short time (< 200 ns) available to MD simulation. In reality experiment, the electric field can be 2 orders of magnitude weaker as the timescale in the laboratory could be on the order of hours.

As an example, the interfacial water can undergo a sudden, reversible phase transition to become ice in electric fields of 10^6 V/m at room temperature²³. The critical field strength for the freezing transition was three order of magnitude weaker than the theoretically predicted value ($> 10^9$ V/m)²⁵ for alignment of water dipoles and crystallization into polar cubic ice.

Lastly, we found a report that an electric field as high as 40 V/nm can be experimentally realizable⁶.

* page 5, line 100: What is the spacegroup of the new ice? This will reflect its polar nature. So it is relevant at this stage of the manuscript.

Response: The space group is FDD2 (see Figure S3).

* page 6, line 127: The rotational degrees of freedom are restricted in all phases of ice due to the hydrogen bonding. The authors need to make their point clearer here.

Response: Thanks for the comment. In the electric field, the dipole of water molecules is forced along the direction of electric field, *i.e.* Z-direction, so the degrees of freedom along X-direction and Y-direction are limited. We have made changes in the revised manuscript

* page 6, line 136: It needs to be stated if the new ice "melts" during the hysteresis test or not. If it melts, then it would not qualify as a ferroelectric phase.

Response: Thanks for the comment. We have performed the hysteresis test, and our simulation results show that the ice χ did not melt during the hysteresis test.

* Figure 3(a): How was the phase diagram constructed? Are the "wiggly" phase boundaries real?

Response: At each given pressure, we set different electric field in our MD simulations. In each simulation, we directly observed liquid-to-solid phase transition in different electric field. And the wiggly line was obtained by connecting all

transition points. It was a real boundary.

* Figure 4: Does it not make sense to only use hydrogen ordered phases for comparison here? Are the authors sure they used ice VI and not ice XV? If they used ice VI, how was the hydrogen disorder implemented and represented?

Response: Note that ice V and VI are hydrogen-disordered. We used Pauli entropy to describe the hydrogen order (or disorder) when calculating free energy.

* page 10, line 241: Only TIP4P was tested for the DFT calculations. Other functionals should be tested as well. It should also be acknowledged that DFT generally favours ferroelectric structure due to the tin-foil approximation. Please see: J. Phys. Chem. Lett. 2014, 5, 4122–4128

Response:

As pointed out by Reviewer #1, the vdW-DFT or rPW86-vdW2 functional is one of the two best for ice computation.

We also add new results with using the second best SCAN functional (see Supplementary results). The results are consistent with vdW-DF2 computations, demonstrating that ice χ is one of the most stable very-high-density ices in the high-pressure region at zero temperature.

Also, the difference between ice B and polar ice B is whether it has an overall net dipole. And the energy of polar ice B is higher than that of ice B as the polar ice B has a net dipole. Most importantly, our free energy calculations by using MC/MD simulations also show that the new ice χ is a stable phase. So the prediction is more general, rather than solely hinges on DFT.

* page 10, line 254: Again, it needs to be acknowledged that the experimental preparation of the new ice would not be possible since the water would chemically decompose into oxygen and hydrogen gas. Also, I am still a bit confused if the authors suggest that the new ice would be stable in absence of an electric field once formed? The phase diagram in Figure 5 seems to suggest that it is thermodynamically stable in the absence of an electric field. However, in the last paragraph it is also said that the new ice is only "dynamically stable". These points are obviously key and need to be spelled out very clearly.

Response: The phonon spectrum indicates that the new ice is dynamically stable, and the free energy calculation indicates it is thermodynamic stable.

* page 16, Table 1: In-situ values for the densities of ices II and VI are available in the literature and should be used here. D. Fortes has published in-situ data for both phases.

Response: For ice II, we noted that D. Fortes did not include the data of average distance between oxygen atoms. The reported values of ice II in D. Fortes' paper were cited from other reference.

Reviewer #3:

Dear Editor

I have read the manuscript titled “Room-temperature electrofreezing of liquid water: a “missing” dense ice phase in the phase diagram” by Zhu and coworkers.

In their work, the authors perform a numerical study of water under an applied high electric field and high pressure and propose a new ice form, ice XX, presented as the most stable structure in the high pressure/low temperature region. According to the authors, ice XX is a polarized ferroelectric phase, whose nucleation occurs only under these extreme conditions.

I regret to state that the manuscript is not well written, missing not only the clearness of the language but also too much information necessary to be able to reproduce the presented results, thus not deserving publication in Nature Communication.

In what follows, I will give few reasons to support my choice.

1- The authors seem to suggest they have discovered a new ice phase, that should appear in the experimental phase diagram under an applied high electric field of few V/nm.

However, they never comment on the fact that this value of the electric field is far beyond the dielectric strength of real water. Therefore, one could argue that this phase would never appear in nature.

Response: Thanks for the comment. We note that in the previous work by Saitta *et al.*, they indicated that electric fields beyond a threshold of about 3.5 V/nm can dissociate water molecules¹¹. So in our work, we adopt an electric field < 3.5 V/nm.

Note also that we use a relatively high electric field in MD simulation to speed up the ice formation due to the short time (< 200 ns) available to MD simulation. In reality experiment, the electric field can be 2 orders of magnitude weaker as the timescale in the laboratory could be on the order of hours.

As an example, the interfacial water can undergo a sudden, reversible phase transition to become ice in electric fields of 10^6 V/m at room temperature²³. The critical field strength for the freezing transition was three order of magnitude weaker than the theoretically predicted value ($> 10^9$ V/m)²⁵ for alignment of water dipoles and crystallization into polar cubic ice.

2- In their study, the authors use TIP4P/2005 water but never discuss the fact that for this water model, the dielectric constants of liquid water and of ice have the opposite behaviour with respect to their real counterparts.

Response: We choose the TIP4P/2005 force field for researching the phase behaviors of water because it is well known to ice-phase community (see Reviewer #1 statement)

that this water force field can describe the relative stability of all ice phases well^{26, 27}. So, the opposite dielectric constants for TIP4P/2005 liquid and ice does not have direct impact to the well-produced ice phase diagram.

There is always some deficiency for any classical model of water, and any model cannot yield all the properties of water nicely. A well-known property is the dipole of water molecule itself, which has to be enhanced artificially to reproduce many condensed phase properties. So one should choose water model carefully to study intended properties (for our work, the phase diagram of ice, which has been validated by many researchers in the field).

It is worth mentioning that we also calculated the stability of the ice phases by the first principle method. The first principle calculation can also support our conclusion very well.

3- The authors vary parameters such as pressure and electric field, not knowing the underlying phase diagram of the water model, and not even properly computing it.

Response: The phase diagrams of several popular water models (TIP3P, TIP4P, TIP5P and TIP4P/2005) have already been reported by Vega *et al.*²⁷ (their figure is cited below), and the TIP4P/2005 water model provides the best description of water property, especially in phase diagram description. Today, most MD work on ice would use one of these models because of the underlying phase diagram is already known. Based on the previously known phase diagram, we could study the effect of electric field on each ice phase in the known phase diagram.

[redacted]

Fig. 5 Phase diagram of TIP4P and TIP4P/2005 models. Symbols: experimental phase diagram, lines: computed phase diagram.

4- The authors study the stability of the novel ice at 0K by means of enthalpy. Next, they establish that this ice should be located between ice II and VI between 0K and 120K. However, when presenting the free-energy calculations, not only the numerical

procedure is not well explained and numerical tables are missing, but also the Pauling entropy is not even mentioned, even though its contribution on the stability of the proposed ice might play an important role and should have been discussed by the authors.

Response: We have considered Pauling entropy according this reference²⁸. The same method has been adopted in our previous work *Sci. Adv.* **2**, e1501010 (2016).²⁹. We have clarified this in the method section of the revised manuscript.

References for the response:

1. Gopalakrishnan, S.; Liu, D.; Allen, H. C.; Kuo, M.; Shultz, M. J., Vibrational spectroscopic studies of aqueous interfaces: salts, acids, bases, and nanodrops, *Chemical reviews* **106**, 1155-75 (2006).
2. Brubach, J. B.; Mermet, A.; Filabozzi, A.; Gerschel, A.; Roy, P., Signatures of the hydrogen bonding in the infrared bands of water, *J. Chem. Phys.* **122**, 184509 (2005).
3. Pruzan, P.; Chervin, J. C.; Canny, B., Stability domain of the ice VIII proton - ordered phase at very high pressure and low temperature, *J. Chem. Phys.* **99**, 9842-9846 (1993).
4. Jorgensen, J. D.; Beyerlein, R. A.; Watanabe, N.; Worlton, T. G., Structure of D2O ice VIII from in situ powder neutron diffraction, *J. Chem. Phys.* **81**, 3211-3214 (1984).
5. Umemoto, K.; Wentzcovitch, R. M., Theoretical study of the isostructural transformation in ice VIII, *Phys. Rev. B* **71**, (2005).
6. Alemani, M.; Peters, M. V.; Hecht, S.; Rieder, K. H.; Moresco, F.; Grill, L., Electric field-induced isomerization of azobenzene by STM, *J Am Chem Soc* **128**, 14446-14447 (2006).
7. Rai, D.; Kulkarni, A. D.; Gejji, S. P.; Pathak, R. K., Water clusters (H₂O)_n, n=6-8, in external electric fields, *J. Chem. Phys.* **128**, 034310 (2008).
8. Choi, Y. C.; Pak, C.; Kim, K. S., Electric field effects on water clusters (n = 3-5): systematic ab initio study of structures, energetics, and transition states, *J. Chem. Phys.* **124**, 94308 (2006).
9. Hermansson, K.; Ojamae, L., On the Role of Electric-Fields for Proton-Transfer in Water, *Solid State Ionics* **77**, 34-42 (1995).
10. Qiu, H.; Guo, W., Electromelting of confined monolayer ice, *Phys. Rev. Lett.* **110**, 195701 (2013).
11. Saitta, A. M.; Saija, F.; Giaquinta, P. V., Ab initio molecular dynamics study of dissociation of water under an electric field, *Phys. Rev. Lett.* **108**, 207801 (2012).
12. Kaneko, T.; Bai, J.; Akimoto, T.; Francisco, J. S.; Yasuoka, K.; Zeng, X. C., Phase behaviors of deeply supercooled bilayer water unseen in bulk water, *Proc. Natl. Acad. Sci. USA* **115**, 4839-4844 (2018).
13. Nomura, K.; Kaneko, T.; Bai, J.; Francisco, J. S.; Yasuoka, K.; Zeng, X. C., Evidence of low-density and high-density liquid phases and isochore end point for water confined to carbon nanotube, *Proc. Natl. Acad. Sci. U. S. A.* **114**, 4066-4071 (2017).
14. Fukazawa, H.; Hoshikawa, A.; Ishii, Y.; Chakoumakos, B. C.; Fernandez-Baca, J. A., Existence of ferroelectric ice in the universe, *Astrophys. J.* **652**, L57-L60 (2006).
15. Fukazawa, H.; Hoshikawa, A.; Chakoumakos, B. C.; Fernandez-Baca, J. A., Existence of ferroelectric ice on planets—A neutron diffraction study, *Nucl. Instrum. Methods Phys. Res. A* **600**, 279-281 (2009).
16. Tribello, G. A.; Slater, B., Proton ordering energetics in ice phases, *Chem. Phys. Lett.* **425**,

246-250 (2006).

17. Singer, S. J.; Kuo, J. L.; Hirsch, T. K.; Knight, C.; Ojamae, L.; Klein, M. L., Hydrogen-bond topology and the ice VII/VIII and ice Ih/XI proton-ordering phase transitions, *Phys. Rev. Lett.* **94**, 135701 (2005).
18. Kamb, B.; Hamilton, W. C.; LaPlaca, S. J.; Prakash, A., Ordered Proton Configuration in Ice II, from Single - Crystal Neutron Diffraction, *J. Chem. Phys.* **55**, 1934-1945 (1971).
19. La Placa, S. J.; Hamilton, W. C.; Kamb, B.; Prakash, A., On a nearly proton - ordered structure for ice IX, *J. Chem. Phys.* **58**, 567-580 (1973).
20. Raza, Z.; Alfe, D.; Salzmann, C. G.; Klimes, J.; Michaelides, A.; Slater, B., Proton ordering in cubic ice and hexagonal ice; a potential new ice phase--XIc, *Phys. Chem. Chem. Phys.* **13**, 19788-95 (2011).
21. Koza, M. M.; Schober, H.; Fischer, H. E.; Hansen, T.; Fujara, F., Kinetics of the high- to low-density amorphous water transition, *J Phys-Condens Mat* **15**, 321-332 (2003).
22. Koza, M. M.; Geil, B.; Winkel, K.; Kohler, C.; Czeschka, F.; Scheuermann, M.; Schober, H.; Hansen, T., Nature of amorphous polymorphism of water, *Phys. Rev. Lett.* **94**, 125506 (2005).
23. Choi, E. M.; Yoon, Y. H.; Lee, S.; Kang, H., Freezing transition of interfacial water at room temperature under electric fields, *Phys. Rev. Lett.* **95**, 085701 (2005).
24. Svishchev, I. M.; Kusalik, P. G., Crystallization of liquid water in a molecular dynamics simulation, *Phys. Rev. Lett.* **73**, 975-978 (1994).
25. Svishchev, I. M.; Kusalik, P. G., Electrofreezing of liquid water: A microscopic perspective, *J. Am. Chem. Soc.* **118**, 649-654 (1996).
26. Aragonés, J. L.; Noya, E. G.; Abascal, J. L.; Vega, C., Properties of ices at 0 K: a test of water models, *J. Chem. Phys.* **127**, 154518 (2007).
27. Vega, C.; Abascal, J. L. F.; Conde, M. M.; Aragonés, J. L., What ice can teach us about water interactions: a critical comparison of the performance of different water models, *Faraday Discuss* **141**, 251-276 (2009).
28. Macdowell, L. G.; Sanz, E.; Vega, C.; Abascal, J. L., Combinatorial entropy and phase diagram of partially ordered ice phases, *J. Chem. Phys.* **121**, 10145-10158 (2004).
29. Huang, Y.; Zhu, C.; Wang, L.; Cao, X.; Su, Y.; Jiang, X.; Meng, S.; Zhao, J.; Zeng, X. C., A new phase diagram of water under negative pressure: The rise of the lowest-density clathrate s-III, *Sci. Adv.* **2**, e1501010 (2016).

Reviewers' comments:

Reviewer #1 (Remarks to the Author):

The revised manuscript addresses mostly convincingly all my initial concerns (I will largely defer to the other reviewers regarding their respective criticisms).

That said, a few points still require further elaboration and/or clarification in my eyes.

Subject to according revision I recommend their to be published.

(1) the stability of ice xi is extremely subtle and at least its thermodynamic stability at zero field is clearly very sensitive to the level of theory at which quantum nuclear motion is accounted for. After all its maximum relative stability with respect to the known phases is reduced from an already small ~ 4 kJ/mol to less than 2 kJ/mol upon inclusion of quantum nuclear motion within the harmonic approximation. Calculations taking into account anharmonic quantum nuclear motion may be beyond the scope of the paper as intended by the authors. However, I feel that this limitation of their current work warrants discussion in the paper.

(2) the datapoints in Figure 2 do not seem to indicate that "when the field orientation is initially along the z-axis and the reversed against the z-axis, a strong hysteresis loop is observed" in that no data for non-zero field in the reversed direction is indicated.

(3) the choice of a 560 molecule simulation cell for the MD simulations evidencing the formation of ice xi seems rather fortuitous and makes me wonder whether the crystal structure of ice xi had previously been obtained/suggested/identified using other methods?

More detailed comments:

(4) when talking of "the first-principles DFT method" it would seem to be clearer to explicitly refer to "DFT with the vdW-DF2 functional".

(5) their general statement that "the electric field induced crystallisation of liquid water can serve as a generic approach to attain new phase-structure of water" seems rather definitive given that (setting aside field induced proton-ordering of e.g. ice I) this is probably the first instance of a new thermodynamically-stable crystal structure of water that is induced by an electric field. I would suggest a more tentative wording.

(6) when stating that "the electric field restricts the rotational degrees of freedom of water molecules" it seems worth to clarify the implications on free energy of water under the electric field and the relative stability of ice xi and water.

(7) can that authors confirm that the 56-molecule cell shown in Figure S4 is not a 2x2x1 supercell of a 14-molecule unit cell? At first glance it seem like this might be the case.

(8) when discussing "Relative stability among various ice polymorphs at 0K based on DFT computation" it might be clearer to explicitly state that what is going to be discussed are relative stability at 0K taking into account quantum nuclear motion within the harmonic approximation.

(9) is the code used to perform the isothermal-isobaric MC simulations publicly available?

(10) why the mismatch in real-space cut-off for Ewald summation between the T-P MD simulations and those at finite electric field?

Prompted by the other reviewers comments I further have the following small comments:

(11) Figure 3(a) would benefit from indication which pressures and temperatures were simulated.

(12) I concur with reviewer 3 that it would make sense to also consider explicitly the proton-ordered counterparts of ices I, II and VI. For example at 0K ice I favours its proton-ordered form, XIh, with Cmc21 symmetry with other realisations of proton-order within the Bernal-Fowler ice rules varying by up to around 6meV/molecule. Thus choosing a particular/or multiple realisations of how the oxygen lattice is dressed with hydrogens may lead to differences of that order subject to the particular choice. Given the subtle stability of ice xi this is significant.

(13) the third reviewer points out that TIP4P/2005 has limitations regarding the description of the dielectric constant of water and ice. In my mind this is relevant to simulations involving electric fields and the fact that TIP4P/2005 performs well in general (e.g. in terms of the T-P phase diagram) when no electric field is applied does not guarantee that it will continue to do so in the presence of strong applied electric fields. In my mind the reviewers criticism has not really been addressed.

Reviewer #2 (Remarks to the Author):

Dear Editor,

As mentioned in my earlier report, I believe the authors present an interesting new finding. However, they have still not fully addressed my concerns (some major) which are listed below.

* I am happy for the authors to maintain "ferroelectric" but I insist that "hydrogen-ordered" is used instead of "proton-ordered". "Proton-ordered" is chemically incorrect.

* Given that the new phase is polar, the authors should really mention and discussed these two papers: J. Phys. Chem. C 2014, 118, 26264–26275 and J. Phys. Chem. Lett. 2014, 5, 4122–4128. The latter is important since it illustrates that polar ice phases are (perhaps artificially) stabilised with computational approaches.

* It is a mistake to take the density classifications from the amorphous ices and apply them to the crystalline phases. I only consider ice VII/VIII to be a very-high-density phase and of course those predicted in the terapascal regime. Ices II, III, V and VI are all rather similar in density and the big jump in density is found for going to ice VII/VIII. The newly predicted phase should be called a high-density phase of ice.

* I need to be clear that I can not recommend publication if the statement is maintained that the new phase of ice could be made experimentally. The computational approaches to prove this are obviously not accurate and the time-scales are very short. The authors cite an experimental paper for a high electric field. In this article, an STM tip was used to generate the high field - this would never enable experimentalist to prepare a new bulk phase. It needs to be stated clearly that ice would not be stable in experiments under the quoted electric fields.

* I still have reservations against the "wiggly" lines in the phase diagram (Figure 3a). Certainly, this can not be real. Perhaps error bars could be included to illustrate this? It would also help if the authors indicated the studied p/T points.

* Regarding the question of ice V vs. XIII and VI vs. XV. I still think that the ordered phases should be included in the discussions and it should be tested if they compete with the new ice at low temperature. I note that ices XIII and XV are not included in the phase diagram (Figure 5). Perhaps ices XIII and XV are more stable than the new ice? I need to add that I am worried about the certainly much too low temperature scale of Figure 5 compared to the experimental situation. The worry is of course that the new ice is only found because the temperature axis is highly

inaccurate.

Reviewer #3 (Remarks to the Author):

I have read the revised version and the comments raised by all three referees.
I acknowledge the fact that the authors have implemented all suggested changes and I am now in the position to recommend the article for publication in Nature Communication.

Reviewers' comments:

Reviewer #3 (Remarks to the Author):

I have read the revised version and the comments raised by all three referees. I acknowledge the fact that the authors have implemented all suggested changes and I am now in the position to recommend the article for publication in Nature Communication.

Response: We thank the reviewer #3 for the very positive recommendation.

Reviewer #1 (Remarks to the Author):

The revised manuscript addresses mostly convincingly all my initial concerns (I will largely defer to the other reviewers regarding their respective criticisms). That said, a few points still require further elaboration and/or clarification in my eyes. Subject to according revision I recommend their to be published.

Response: We thank the Reviewer #1 for the positive comments.

(1) the stability of ice xi is extremely subtle and at least its thermodynamic stability at zero field is clearly very sensitive to the level of theory at which quantum nuclear motion is accounted for. After all its maximum relative stability with respect to the known phases is reduced from an already small ~ 4 kJ/mol to less than 2 kJ/mol upon inclusion of quantum nuclear motion within the harmonic approximation. Calculations taking into account anharmonic quantum nuclear motion may be beyond the scope of the paper as intended by the authors. However, I feel that this limitation of their current work warrants discussion in the paper.

Response: Thanks for the insightful comment and suggestion. We add a discussion on this in the revised manuscript (also add new calculations and Figure S7). Besides the Figure S6 where we show the enthalpy per water molecule based on SCAN functional calculations, we also perform new computation using the state-of-the-art functional vdW-DF2 (see Figure S7).

Most importantly, we find that the maximum relative enthalpy difference of the ices is only 0.25 kJ/mol between the free energy with and without the zero-point energy (ZPE) correction, a very small value. In other words, our results indicate that the ZPE correction (reflecting quantum nuclear motion to some extent) has little effect on the relative stability of the ices considered. ZPE correction only changes slightly the location of phase transition pressure. We agree with the reviewer that calculations with including effects of anharmonic quantum nuclear motion is a challenging computational task. Since the ZPE correction at harmonic level is very small, we expect the correction due to anharmonic quantum nuclear motion may only slightly change the location of phase transition pressure as well, but unlikely to dramatically change the relative stability.

Figure S7. Relative enthalpy per water molecule (calculated based on vdw-DF2 functional) *versus* P for ice χ , ice II, ice XI, ice B, and polar ice B, where ice VI is taken as the reference: (a) Without ZPE, and (b) with ZPE correction.

(2) the datapoints in Figure 2 do not seem to indicate that “when the field orientation is initially along the z-axis and the reversed against the z-axis, a strong hysteresis loop is observed” in that no data for non-zero field in the reversed direction is indicated.

Response: A new figure caption is added in Figure 2, to explain the data points clearer.

(3) the choice of a 560 molecule simulation cell for the MD simulations evidencing the formation of ice xi seems rather fortuitous and makes me wonder whether the crystal structure of ice xi had previously been obtained/suggested/identified using other methods?

Response: We should point out that our simulation results are not sensitively dependent on the number of water molecules. Actually, we have also obtained the same ice xi in other independent simulation, e.g., for system with 500 water molecules, and 2000 water molecules. Because the spontaneous formation of ice χ requires not only specific high pressure range but also ultra-high electric field, we think this is why ice χ has not been seen in previous simulation/experimental work. Like many cases, the discovery was serendipitous.

More detailed comments:

(4) when talking of “the first-principles DFT method” it would seem to be clearer to explicitly refer to “DFT with the vdW-DF2 functional”.

Response: Thanks for the suggestion. We have made all changes in revised manuscript.

(5) their general statement that “the electric field induced crystallisation of liquid water can serve as a generic approach to attain new phase-structure of water” seems rather definitive given that (setting aside field induced proton-ordering of e.g. ice I) this is probably the first instance of a new thermodynamically-stable crystal structure of water that is induced by an electric field. I would suggest a more tentative wording.

Response: Thanks for the comment. We have made the change “The electric-field induced crystallization of liquid water can serve as an alternative approach to attain new phase structures of water, particularly the ferroelectric ices.”.

(6) when stating that “the electric field restricts the rotational degrees of freedom of water molecules” is seems worth to clarify the implications on free energy of water under the electric field and the relative stability of ice xi and water.

Response: Thanks for the comment. Under the electric field, the dipole orientations of water molecules tend to be along the direction of electric field while the water molecules can still adapt to form the hydrogen-bonding network. Their interplay could lead to a different and yet more stable solid state. We clarified the unclear expression in our revised manuscript.

(7) can that authors confirm that the 56-molecule cell shown in Figure S4 is not a 2x2x1 supercell of a 14-molecule unit cell? At first glance it seem like this might be the case.

Response: Thanks for the comment. The unit cell of 56 water molecules reported in the manuscript is a conventional cell, and the corresponding primitive cell has only 14 water molecules. The lattice parameters of the primitive cell are $a = 6.63 \text{ \AA}$, $b = 12.36 \text{ \AA}$, $c = 13.68 \text{ \AA}$, $\alpha = 28.90^\circ$, $\beta = 64.36^\circ$ and $\gamma = 86.74^\circ$. To better illustrate the structural feature, the 56-water molecules cell was selected as the unit cell. We added the structure of primitive cell of ice χ in Figure S4 (e).

(8) when discussing “Relative stability among various ice polymorphs at 0K based on DFT computation” it might be clearer to explicitly state that what is going to be discussed are relative stability at 0 K taking into account quantum nuclear motion within the harmonic approximation.

Response: Thanks for the valuable suggestion. We clarified this unclear expression in our revised manuscript (see Page 9).

(9) is the code used to perform the isothermal-isobaric MC simulations publicly available?

Response: This code is our home-made program for performing the isothermal-isobaric MC simulations. Based on this code, we have successful reproduced the phase diagram (see our previous papers^{1,2}). The code is not publicly available right now, but we are

willing to share it with collaborating groups.

(10) why the mismatch in real-space cut-off for Ewald summation between the T-P MD simulations and those at finite electric field?

Response: We note that the free-energy calculations are computationally very expensive. Since the simulation system must be the supercell of ice phases, we need to carefully adopt appropriate sizes. The minimum length of the systems in our simulations is about 19 Å, and the value of cutoff must be less than half of the minimum length of the box. Therefore, the cutoff of 8.5 Å is used in our free-energy calculations (this cutoff 8.5 Å has been widely used in previous studies of ice phases^{3,4}; and we can directly compare our results with those reported previously). On the other hand, the MD simulation for liquid water at finite electric field is less expensive. So we can afford larger real-space cutoff.

Prompted by the other reviewers comments I further have the following small comments:

(11) Figure 3(a) would benefit from indication which pressures and temperatures were simulated.

Response: We agree to referee's comment. A legend is added in Figure 3(a) to indicate the temperature is at 270 K while at numerous pressures (P) ranging from 0.001 kbar to 10 kbar (see Page 6).

(12) I concur with reviewer 3 that it would make sense to also consider explicitly the proton-ordered counterparts of ices I, II and VI. For example at 0K ice I favours its proton-ordered form, XIh, with Cmc21 symmetry with other realisations of proton-order within the Bernal-Fowler ice rules varying by up to around 6 meV/molecule. Thus choosing a particular/or multiple realisations of how the oxygen lattice is dressed with hydrogens may lead to differences of that order subject to the particular choice. Given the subtle stability of ice xi this is significant.

Response: Thanks for the valuable suggestion. If we suppose that ice χ is a hydrogen-disordered ice phase, based on the same oxygen atom occupancies with ice χ , nine structures with different orientations of hydrogens are generated, which are shown in new Figure S8 (see below). To compare the relative stability of ferroelectric ice χ with these nine newly generated ice structures, we calculated their enthalpies under different pressures at 0 K. As the new Figure S9 (see below) shows, ice χ is still the most stable structure among them, although their enthalpy differences are very small (less than 0.4 kJ/mol). We hope future experiments can ultimately determine whether the new ice is proton-ordered or hydrogen-disordered.

Figure S8. Top view and side view of nine hydrogen-disordered ice χ structures. Oxygen atoms are depicted as red balls, hydrogen atoms as white balls, and hydrogen bonds as blue dotted lines.

Figure S9. Relative enthalpy per water molecule (based on vdw-DF2 calculations) versus pressure for ice χ and the nine randomly generated hydrogen-disordered ice structures, where ice χ is taken as the reference.

(13) the third reviewer points out that TIP4P/2005 has limitations regarding the description of the dielectric constant of water and ice. In my mind this is relevant to simulations involving electric fields and the fact that TIP4P/2005 performs well in general (e.g. in terms of the T-P phase diagram) when no electric field is applied does not guarantee that it will continue to do so in the presence of strong applied electric fields. In my mind the reviewers criticism has not really been addressed.

Response: Thanks for the comment. We note that the ice χ reported here is insensitive to the water model. Actually, we find that the ice χ can also be obtained by using three other water model potentials, such as SPC/E, TIP4P and TIP5P (see Table R1), although there are minor differences among simulation results from different water models. Note that the dielectric constant of SPC/E and TIP5P water model are 71 and 81.5 respectively, both being very close to that of experimental result (78.4). So these independent simulations with different water models give more credence to the new ice χ phase.

Table R1 ice χ is obtained with various water models at different T-P-E conditions.

Water model	P (kbar)	T (K)	E_z (V/nm)
SPC/E	10.0	245	2.0
TIP4P	8.0	260	2.5
TIP5P	10.0	300	3.0

Reviewer #2 (Remarks to the Author):

As mentioned in my earlier report, I believe the authors present an interesting new finding. However, they have still not fully addressed my concerns (some major) which are listed below.

* I am happy for the authors to maintain "ferroelectric" but I insist that "hydrogen-ordered" is used instead of "proton-ordered". "Proton-ordered" is chemically incorrect.

Response: This point is well taken now and we have changed all relevant terms to the "hydrogen-ordered" in our revised manuscript.

* Given that the new phase is polar, the authors should really mention and discussed these two papers: J. Phys. Chem. C 2014, 118, 26264–26275 and J. Phys. Chem. Lett. 2014, 5, 4122–4128. The latter is important since it illustrates that polar ice phases are (perhaps artificially) stabilised with computational approaches.

Response: We agree with this point and add the following discussion in Introduction and especially mentioned Del Ben et al. work:

The ice XV, the hydrogen-ordered form of ice VI phase, is antiferroelectric (P1) according to experimental observation¹, whereas it is predicted to be a ferroelectric Cc hydrogen-ordered structure based on local density-functional approach.²⁷⁻²⁸ However, Del Ben et al. used high-level *ab initio* computation and predicted that the antiferroelectric phase is indeed the ground state²⁹, suggesting that more accurate density-function approaches should be considered (see below).

* It is a mistake to take the density classifications from the amorphous ices and apply them to the crystalline phases. I only consider ice VII/VIII to be a very-high-density phase and of course those predicted in the terapascal regime. Ices II, III, V and VI are all rather similar in density and the big jump in density is found for going to ice VII/VIII. The newly predicted phase should be called a high-density phase of ice.

Response: Thanks for the suggestion. We have changed the classifications of ice phases in our revised manuscript, and call this new ice as a high-density ice.

* I need to be clear that I can not recommend publication if the statement is maintained that the new phase of ice could be made experimentally. The computational approaches to prove this are obviously not accurate and the time-scales are very short. The authors cite an experimental paper for a high electric field. In this article, an STM tip was used to generate the high field - this would never enable experimentalist to prepare a new bulk phase. It needs to be stated clearly that ice would not be stable in experiments under the quoted electric fields.

Response: In our previous response letter, we pointed out that the electric field can be 2 orders of magnitude lower because the timescale in the laboratory can be on the order of hours in laboratory (11 orders of magnitude longer in time-scales than simulation). We cannot be for sure to say that ice would not be stable in experiments under the quoted electric fields. But to comply with the reviewer #2's point, in the end of conclusion we add "In light of the requirement of ultra high electric field, whether this predicted ice χ can be produced in the laboratory via electrofreezing of liquid water remains to be an open question."

* I still have reservations against the "wiggly" lines in the phase diagram (Figure 3a). Certainly, this cannot be real. Perhaps error bars could be included to illustrate this? It would also help if the authors indicated the studied p/T points.

Response: Thanks for the suggestion. In Figure 3a, the electric field strength is varied from 0 to 3.5 V/nm by an increment of 0.1 V/nm, thus, the error bar of the electric field strength is 0.05 V/nm. We add the error bar state in caption of Figure 3a (see Page 21).

* Regarding the question of ice V vs. XIII and VI vs. XV. I still think that the ordered phases should be included in the discussions and it should be tested if they compete with the new ice at low temperature. I note that ices XIII and XV are not included in the phase diagram (Figure 5). Perhaps ices XIII and XV are more stable than the new ice? I need to add that I am worried about the certainly much too low temperature scale of Figure 5 compared to the experimental situation. The worry is of course that the new ice is only found because the temperature axis is highly inaccurate.

Response: Thanks for the comment. Three ice phases next to ice χ are ice II, ice V and ice VI. Ice XIII is the metastable proton-ordered form of ice V formed by doping with 10 mM HCl below 130 K at 500 MPa, while Ice XV is the proton-ordered form of ice VI, which is thermodynamically stable at temperature below ~ 130 K^{5,6}. By contrast, ice II has an ordered arrangement of hydrogen bonds, without having a disordered form.

To address the comment, we have considered relative stability of ice χ , ice XIII and ice XV in our calculations. Specifically, three temperatures, i.e., 0 K, 40 K and 120 K, are considered. For each temperature, two pressures, i.e., 8.0 kbar and 12.0 kbar, are considered. Under these T-P conditions, the T-P phase diagram for TIP4P/2005 water model shows that ice χ is the most stable structure with ice XIII and ice XV being included. Table R2 gives free-energy calculation results, which show that ice χ is still the most stable among the three ice phases under these conditions. We hope these new results remove the worry of reviewer #2.

Table R2. Free energy of ice XIII, ice XV and ice χ at three different temperatures (T) and two different pressures (P). Free energies are in $kcal \cdot mol^{-1}$ units. Numbers in brackets represent the number of water molecules in our simulation system (N).

T (K)	P (kbar)	Free Energy per water molecule ($G/kcal \cdot mol^{-1}$)		
		Ice XIII(336)	Ice XV (640)	Ice χ (448)
0	8.0	-11.843	-11.978	-12.168
	12.0	-10.523	-10.770	-10.936
40	8.0	-2.229	-2.285	-2.476
	12.0	-0.906	-1.073	-1.236
120	8.0	-4.797	-4.749	-4.908
	12.0	-3.469	-3.517	-3.648

References

- Huang, Y.; Zhu, C.; Wang, L.; Cao, X.; Su, Y.; Jiang, X.; Meng, S.; Zhao, J.; Zeng, X. C., A new phase diagram of water under negative pressure: The rise of the lowest-density clathrate s-III, *Sci. Adv.* **2**, e1501010 (2016).
- Huang, Y.; Zhu, C.; Wang, L.; Zhao, J.; Zeng, X. C., Prediction of a new ice clathrate with record low density: A potential candidate as ice XIX in guest-free form, *Chem. Phys. Lett.* **671**, 186-191 (2017).
- Conde, M. M.; Gonzalez, M. A.; Abascal, J. L.; Vega, C., Determining the phase

diagram of water from direct coexistence simulations: the phase diagram of the TIP4P/2005 model revisited, *J. Chem. Phys.* **139**, 154505 (2013).

4. Aragoes, J. L.; Conde, M. M.; Noya, E. G.; Vega, C., The phase diagram of water at high pressures as obtained by computer simulations of the TIP4P/2005 model: the appearance of a plastic crystal phase, *Phys. Chem. Chem. Phys.* **11**, 543-555 (2009).

5. Salzmann, C. G.; Radaelli, P. G.; Hallbrucker, A.; Mayer, E.; Finney, J. L., The preparation and structures of hydrogen ordered phases of ice, *Science* **311**, 1758-1761 (2006).

6. Salzmann, C. G.; Radaelli, P. G.; Mayer, E.; Finney, J. L., Ice XV: a new thermodynamically stable phase of ice, *Phys. Rev. Lett.* **103**, 105701 (2009).

REVIEWERS' COMMENTS:

Reviewer #2 (Remarks to the Author):

Dear Editor,

I am essentially happy with this version of the manuscript. However, two points that still need to be addressed are: (1) in lines 58/59, there is still talk of "protons" and (2) the fact that the new ice is more stable than ices XIII and XV is something that needs to be mentioned in the manuscript.

Reviewer #3 (Remarks to the Author):

Dear Editor

I have read the reports by the 3 referees and the corresponding replies by the authors. I acknowledge their effort to properly address all issues raised by the referees. For this reason, I recommend the manuscript for publication.

REVIEWERS' COMMENTS:

Reviewer #2 (Remarks to the Author):

Dear Editor,

I am essentially happy with this version of the manuscript. However, two points that still need to be addressed are: (1) in lines 58/59, there is still talk of "protons" and (2) the fact that the new ice is more stable than ices XIII and XV is something that needs to be mentioned in the manuscript.

Response: We thank the reviewer #2 for the positive recommendation.

(1) We have changed the words "proton/protons" to the "hydrogen/hydrogens" in our revised manuscript.

(2) We have added the sentence "Note that the free energy calculations show that ice χ has the lowest free energy among ice χ , ice XIII and ice XV in the low temperature and high pressure region, which indicates that ice χ is more stable than ices XIII and XV." In our revised manuscript.

Reviewer #3 (Remarks to the Author):

Dear Editor

I have read the reports by the 3 referees and the corresponding replies by the authors. I acknowledge their effort to properly address all issues raised by the referees. For this reason, I recommend the manuscript for publication.

Response: We thank the reviewer #3 for the very positive recommendation.